# Interaction of clinical-stage antibodies with heme predicts their physiochemical and binding qualities

Maxime Lecerf [1], Alexia Kanyavuz[1], Sofia Rossini[1] & Jordan D. Dimitrov [1✉]

Immunoglobulin repertoires contain a fraction of antibodies that recognize low molecular weight compounds, including some enzymes' cofactors, such as heme. Here, by using a set of 113 samples with variable region sequences matching clinical-stage antibodies, we demonstrated that a considerable number of these antibodies interact with heme. Antibodies that interact with heme possess specific sequence traits of their antigen-binding regions. Moreover they manifest particular physicochemical and functional qualities i.e. increased hydrophobicity, higher propensity of self-binding, higher intrinsic polyreactivity and reduced expression yields. Thus, interaction with heme is a strong predictor of different molecular and functional qualities of antibodies. Notably, these qualities are of high importance for therapeutic antibodies, as their presence was associated with failure of drug candidates to reach clinic. Our study reveled an important facet of information about relationship sequence-function in antibodies. It also offers a convenient tool for detection of liabilities of therapeutic antibodies.

[1] Centre de Recherche des Cordeliers, INSERM, Sorbonne Université, Université de Paris, F-75006 Paris, France. ✉email: jordan.dimitrov@sorbonne-universite.fr

Antibodies (Abs) represent an essential element of the adaptive immune defense. Typically they recognize unique molecular motifs (epitopes) displayed on biological macromolecules, such as proteins and carbohydrates. Nevertheless, immunoglobulin (Ig) repertoires contain also Abs that bind to various low molecular weight molecules. Thus, pioneering works with human and mouse myeloma-derived monoclonal Abs revealed unexpected high frequencies of molecules that interact with nitroarenes compounds, especially 2,4-dinitrophenyl[1–3]. Further studies showed that the human Abs can also recognize other aromatic and heterocyclic molecules, including cofactors or prosthetic groups exploited by enzymes, namely—riboflavin, FAD, ATP, cobalamin, protoporphyrin IX, and heme[4–14]. Human Abs also recognize a xenogenic disaccharide molecule, i.e., galactosyl-(1-3)-galactose (α-Gal)[15–17]. These studies also suggested that the binding of cofactors or other low molecular weight molecules occurs in the variable regions of Abs. Of note, the interaction of Abs with some cofactors results in a dramatic effect on their functions. Thus, upon contact with heme some Abs acquire reactivity towards large panels of previously unrecognized antigens, i.e., they become polyreactive[10,14,18–20].

Detection of Abs interacting with low molecular weight compounds in human immune repertoires is not anecdotic but these Abs represent a considerable fraction of circulating Igs. Thus, it was estimated that >10% of Abs interact with heme[20,21]; the frequency of Abs recognizing DNP in human repertoires is ca. 1%[2,22], and those recognizing, α-Gal is 1–8%[22,23].

The particular physicochemical properties of heterocyclic molecules can explain the high reactivity with Abs. For example, heme is a complex of tetrapyrrole macrocyclic compound protoporphyrin IX with iron ion (Fig. 1a). The heme molecule displays different types of chemical groups offering the possibility for establishing non-covalent interactions of various nature—hydrophobic, π-electrostatic, ionic bonding, metal coordination, and hydrogen bonding. Not surprisingly heme is a very promiscuous molecule; apart from proteins that use heme constantly as a prosthetic group (hemoproteins), a large number of proteins bind heme in a less stable or transient manner[24,25]. Moreover, heme was reported to interact with lipids, nucleic acids, peptides, and other aromatic compounds[26–28]. Large-scale investigations of the architecture of the heme-binding sites of proteins revealed some clear-cut tendencies. Thus, the binding sites for heme are predominantly enriched in certain amino acids such as histidine, cysteine, methionine, tyrosine, and amino acids with basic side chains[29–31]. All these amino acids can establish direct interactions with different parts of the heme molecule.

The fact that a significant portion of human Ab molecules interacts with heme implies that these Abs may have common molecular features of their binding sites that are suitable for the accommodation of the cofactor molecules. We hypothesize that that interaction with heme can inform for qualities of antigen-binding sites that are expressed further as particular physico-chemical and functional behavior of the Abs. To test this hypothesis, we used a repertoire of clinical-stage therapeutic monoclonal Abs. These Abs were previously extensively and meticulously characterized by a battery of 12 methods that assess different physicochemical and functional properties[32]. Our results obtained with 113 samples with V region sequences

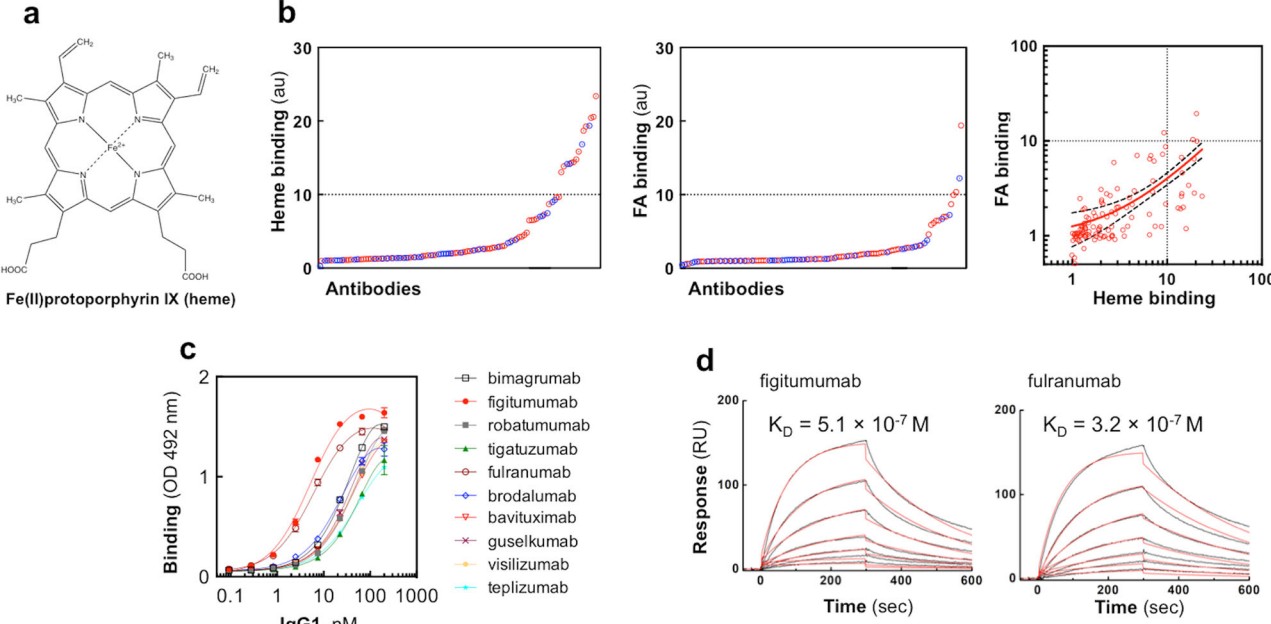

**Fig. 1 Interaction of heme with therapeutic Abs. a** Structural formula of Fe-protoporphyrin IX (heme). **b** Interaction of the repertoire of monoclonal IgG1 therapeutic antibodies to immobilized heme (left panel) and FA (right panel). Each circle on the panels represents the average binding intensity of a given therapeutic Ab analyzed in duplicate. The binding intensity for each Ab is obtained by division of reactivity to cofactor-conjugated gelatine by the reactivity of Ab to gelatine alone. The dashed line represents the tenfold increased binding over binding to gelatine alone. The right panel depicts the correlation of the binding to FA versus the binding to heme. The red circles represent Abs that are currently in clinical trials. The blue circles indicated the clinically approved Abs. The Spearman correlation analyses $\rho = 0.69$, $P < 0.0001$—indicates a significant correlation between the two parameters. The dashed lines display the 95%, confidence band. **c** Concentration-dependent binding of Abs to immobilized heme. The selected ten Abs were identified as top binders in the high-throughput heme-binding assay (panel **b**). Increasing concentrations 0.09–200 nM of the selected Abs were incubated with a heme-coated surface or with the control surface. Nonlinear regression analyses were performed by GraphPad Prism v.6 software. Each circle on the panels represents the average binding intensity ±SD of a given therapeutic Ab analyzed in duplicate. **d** Real-time interaction profiles of binding of heme to immobilized Abs (figitumumab and fulranumab). The black line depicts the binding profiles obtained after injection of serial dilutions of hemin (1250–19.5 nM). The red lines depict the fits of data obtained by global analysis using the Langmuir kinetic model.

corresponding to the clinical-stage therapeutic monoclonal Abs expressed on the human IgG1 framework, revealed that interaction with heme correlated with specific sequence characteristics of the antigen-binding site. Notably, the interaction with heme could be used as a potent predictive surrogate for a number of molecular and functional qualities of Abs, i.e., hydrophobicity, stability, natural polyreactivity, as well as propensities for self-binding and cross-reactivity with other Abs. Thus, in addition to providing evidence for the molecular organization of the paratopes of heme-binding Abs, this study also points to the utility of heme for the prediction of the physicochemical features of the V region and the functional behavior of Abs. These data might extend our basic comprehension of the Ab molecule and pave the way for the development of convenient analytical assays for the early detection of some unwanted properties of candidate therapeutic Abs.

## Results

**Interaction of therapeutic Abs with heme**. Previous studies have demonstrated that a significant fraction of human IgG interacts with heme (Fe-protoporphyrin IX, Fig. 1a)[10]. To examine whether interaction with heme reflects certain physicochemical and functional qualities of Abs we used a repertoire of 113 monoclonal therapeutic Abs all expressed as human IgG1. At present, 44 of those Abs are approved for clinical use, and the rest of the molecules reached at least Phase II or III clinical trials (Supplementary Data 1). These Abs were characterized thoroughly by diverse assays measuring physiochemical and binding properties[32].

First, to assess the binding of Abs to heme, we used a method based on immune sorbent assay, where the oxidized form of heme is covalently attached to a precoated carrier protein. The obtained results clearly demonstrate that a significant portion of Abs bind to immobilized heme (14% of Abs binding with tenfold higher intensity to immobilized heme as compared to carrier alone) (Fig. 1b). Next, the interaction of the therapeutic Abs set with another aromatic cofactor molecule, folic acid (folate), was evaluated. The number of Abs binding to surface-immobilized folate was considerably lower as compared to heme-binding Abs (Fig. 1b). Nevertheless, Spearman's correlation analyses demonstrated a significant correlation between the reactivities of Abs against both heterocyclic cofactors ($\rho = 0.69$, $P < 0.0001$). The Abs that recognized heme with the highest intensity were subjected to additional analysis. Figure 1c depicts the Ab-concentration-dependent binding to immobilized heme. Data indicates that those Abs bind with high avidity to heme (50% of the maximal binding for most of the Abs was achieved by <100 nM of IgG1). Two Abs (figitumumab and fulranumab) recognized surface-immobilized heme with ca. tenfold higher avidity as compared to other Abs. To determine the binding affinity of immobilized Abs to soluble heme, we used surface plasmon resonance (SPR)-based technology. The obtained real-time interaction data of binding of heme to most of the selected heme-binding monoclonal Abs characterized an interaction with $K_D$ values from 1.7 to $5.3 \times 10^{-7}$ M (Fig. 1d and Supplementary Fig. 1). Robatumumab and Brodalumab bound heme with lower affinities, i.e., with $K_D$ values of $1.6 \times 10^{-6}$ and $3.7 \times 10^{-6}$ M, respectively (Supplementary Fig. 1). A possible explanation for lower affinity (especially in the case of Brodalumab) might be a perturbation of Ab structure due to covalent immobilization on the sensor chip.

In addition to the approach based on surface immobilization of heme (or Abs), we analyzed the interaction of heme with Abs in solution by high-throughput UV–vis absorbance spectroscopy, and by examining the changes in the catalytic activity of heme

(Fig. 2). These techniques confirmed that a fraction from the panel of therapeutic monoclonal Abs is capable of binding to heme. Indeed, the increase of absorption intensity in the Soret region of heme expressed as $\Delta A_{\max}$ (Fig. 2b), usually occur as a result of a change of molecular environment of the oxidized form of heme from more polar, typical for the aqueous solution to more hydrophobic, typical for the binding site on the protein. The shift of the wavelength of the absorbance maximum in the Soret region expressed as $\lambda$ max shift (Fig. 2c), indicates interaction of amino acid residues with heme's iron, thus also proving interaction of heme with Ab. Indication for binding of heme to Abs is also changed in the intrinsic peroxidase activity of the former (Fig. 2d).

**Acquisition of antigen-binding polyreactivity after interaction of therapeutic Abs with heme**. To analyze whether the exposure to heme affects the antigen-binding properties of Abs, we next investigated the interaction of the panel of therapeutic Abs with a panel of unrelated proteins—human Factor VIII, human C3, and LysM domains of AtlA from Enterococcus faecalis. The binding of all Abs to immobilized proteins was assessed before and after the exposure to heme. Figure 3a illustrates that in their native state some of the therapeutic Abs have the capacity to recognize the studied proteins. These Abs displayed natural antigen-binding polyreactivity since none of the proteins is a cognate antigen of the studied therapeutic Abs. The presence of polyreactive Abs in the panel of therapeutic monoclonal Abs was also detected by three different assays in a previous study[32]. The obtained results also demonstrated that a fraction of therapeutic Abs acquired the capacity to recognize the proteins following contact with heme (Fig. 3a). The heme-induced Ab reactivity to a given protein strongly correlated with binding to the other two proteins and to average binding reactivity against the three proteins (Fig. 3b). This result implies that the heme-sensitive Abs acquired antigen-binding polyreactivity upon exposure to heme. The Abs that demonstrated the highest gain in reactivity, as assessed by enzyme-linked immunosorbent assay (ELISA), were further subjected to immunoblot analyses. The observed recognition of multiple proteins from bacterial lysate confirmed the ability of heme to transform the binding behavior of certain Abs (Fig. 3c). Previous works suggested that heme endows Abs with polyreactivity by binding to the antigen-binding site and serving as interfacial promiscuous cofactor facilitating contact with unrelated proteins[10,14,33]. We applied computational Ab modeling (Rosetta algorithm) to generate the most probable structural configuration of the antigen-binding site of a heme-binding Ab. The selected Ab (sample corresponding to the sequences of tigatuzumab) displayed the highest sensitivity to heme in terms of both binding and acquisition of polyreactivity (Fig. 3a). By using a ligand-docking algorithm, we predicted the most probable position of the macrocyclic system of heme on the variable region. As can be seen in Fig. 3d, the putative binding site for heme is overlapping with the central part of the antigen-binding region. This model suggests that heme molecule could establish interaction with complementarity-determining region (CDR) loops of both light and heavy Ig chains. The docking analyses of the Ab displaying the highest sensitivity to heme are in full accordance with previous findings that heme binding to the antigen-binding site can be used as an interfacial cofactor for extension of antigen-recognition activity.

Heme influenced selectively only Fab-dependent functions of Abs. Thus, SPR-based analyses revealed that the binding of native and heme-exposed Abs to human neonatal Fc receptor (FcRn) was characterized with no significant differences in the binding affinity (Fig. 4). For these analyses we treated a fixed

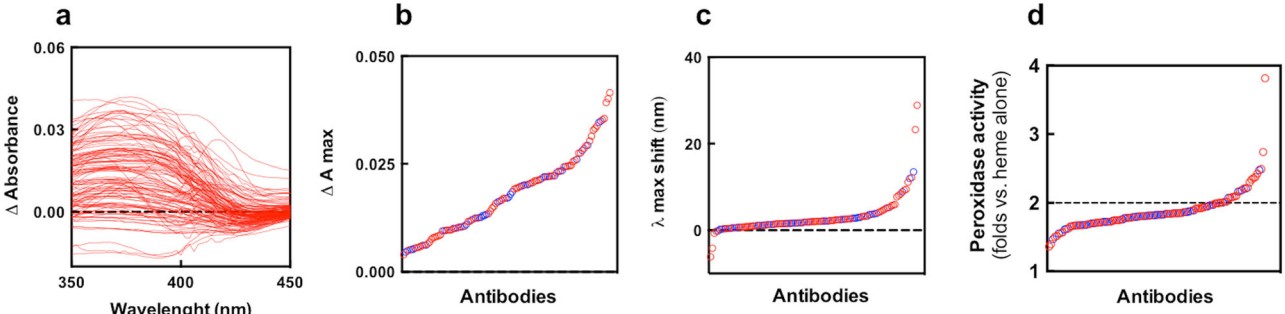

**Fig. 2 Binding of heme to Abs in solution.** UV–vis absorbance spectroscopy analysis of the interaction of repertoire of therapeutic Abs with hemin. **a** Differential absorbance spectra in the wavelength range 350–450 nm obtained after subtraction of the spectrum of 5 μM hemin in PBS from the spectrum of 5 μM hemin in the presence of 670 nM of monoclonal IgG1. Each line depicts the differential spectrum of an individual Ab. **b** Plot showing the maximal increase of the absorbance intensity of heme in the presence of therapeutic Ab. **c** Plot of the value of the shift of the maximal wavelength in the presence of antibody as compared to the absorbance maxima of heme in PBS only. On panels **b** and **c**, each circle depicts the absorbance maximal intensity or shift in the wavelength maxima of individual Ab. **d** Changes of catalytic activity of heme upon interaction with Abs. The oxidation of ABTS was used as a colorimetric reaction for the assessment of the catalytic activity of heme in the absence and presence of monoclonal IgG1. The panel illustrates the fold change in the peroxidase activity, as compared to heme alone in PBS, for the repertoire of therapeutic Abs. Each circle depicts the average changes in the peroxidase activity of an individual monoclonal Ab from duplicate readings. The dashed line indicates a twofold increase in the catalytic activity of heme in the presence of Abs as compared to heme alone. On panels **a**, **c**, and **d** the red circles represent Abs that are currently in clinical trials. The blue circles indicate the clinically approved Abs.

concentration of IgG1 (670 nM) with a tenfold excess of heme, thus ensuring full saturation of heme-binding sites on Abs.

**Sequence characteristics of the antigen-binding site of Abs interacting with heme.** To provide understanding about the molecular features of Abs responsible for interaction with heme as well as the acquisition of binding polyreactivity, we elucidated sequence characteristics of the antigen-binding site of 113 Abs that overlap with molecules previously described in ref. [32]. We focused our analyses on the most diverse regions of the antigen-binding site, i.e., CDR3 of heavy and light chains, as well as CDR2 of the heavy chain, since the latter is the second most diverse region in antigen-binding site after CDRH3 and provides the second largest molecular surface for contact with protein antigens[34]. By applying Spearman correlation analyses, it was observed that the length of CDRH3, CDRH2, and CDRL3 did not correlate with binding to heme or folate or with heme-induced polyreactivity (Fig. 5). No significant correlation was observed with the number of somatic mutations in the V regions either.

The analysis of frequencies of amino acid residues in CDRs, however, revealed a number of statistically significant correlations. Thus, the ability of Abs to recognize heme positively correlated with the presence of a higher number of basic amino acid residues in CDRH3, particularly with the higher number of Lys (Fig. 5). The positive correlation of binding to heme was also observed for the number of Arg residues in CDRH2. This analysis also showed that both heme- and folate-binding Abs had a significantly decreased frequency of acidic residues (Asp and Glu) in CDRH2. The heme-binding Abs have a significantly higher number of Tyr residues in CDRH3. Frequency of another aromatic residue—Phe in CDRH3 and CDRH2 negatively correlated with higher binding to immobilized heme and folate. Interestingly, no statistically significant correlation was observed between the frequency of any amino acid residue type in CDRL3 and the binding to heme or folate. This could be explained by a lower degree of sequence diversity and the significantly smaller average length of the CDR L3 loop as compared with CDR H3.

Next, we investigated the correlation of the heme-mediated acquisition of polyreactivity and sequence features of the antigen-binding site of Abs. The heme-induced polyreactivity of the

therapeutic Abs negatively correlated with the presence of acidic residues in CDRH3. Similarly, as in the case of binding to heme, the Abs that acquired polyreactivity had a significantly higher number of Tyr residues in CDRH3. Heme-sensitive Abs also presented an elevated frequency of Tyr in CDRH2. Abs acquiring polyreactivity following exposure to heme were characterized with lower frequencies of Ala and Leu in CDRH2 and CDRH3, respectively.

Structural models of the selected Abs with the most pronounced heme-binding capacity and/or heme-induced polyreactivity revealed that positive charges and Tyr residues are well displayed on the molecular surface of the antigen-binding site (Fig. 6). The surface accessibility of positively charged amino acid residues and Tyr on the molecular surface of antigen-binding sites and their increased frequency suggest that these residues might play a direct role in contacts with heme's carboxyl groups and aromatic system, respectively. Tyr residues are also known to establish metal–coordination interaction with heme's iron ion.

**Biophysical properties of the therapeutic Abs that interact with heme.** In a previous study, Jain et al.[32] quantified different physicochemical properties of a repertoire of 137 monoclonal therapeutic Abs. The measured features of Abs include—expression yield in HEK cells (HEKt), thermodynamic stability (assessed as melting temperature, Tm), accelerated stability (AS, assessed as long-term storage stability at elevated temperature, i.e., 40 °C), tendency for self-association (ACSIN assay, which is based on gold nanopartilces, and CSI assay—based on biolayer interferometry, for measuring propensity for homotropic binding of Abs), binding to polyclonal human IgG (CIC, assay where the binding of the Abs to sepharose-immobilized pooled IgG was estimated), antigen-binding polyreactivity (PSR assay, where the reactivity of Abs to soluble membrane proteins from CHO cells was assessed in solution; ELISA assay where the reactivity of Abs to surface immobilized antigens—KLH, LPS, ssDNA, dsDNA, and insulin was measured, and BVP assay where binding of Abs to baculovirus particles was evaluated by ELISA) and hydrophobicity (SGAC100 assay where hydrophobicity of Abs was studied by salting-out effect of ammonium sulfate, HIC and SMAC assays, where tendency of Abs for binding to two types of

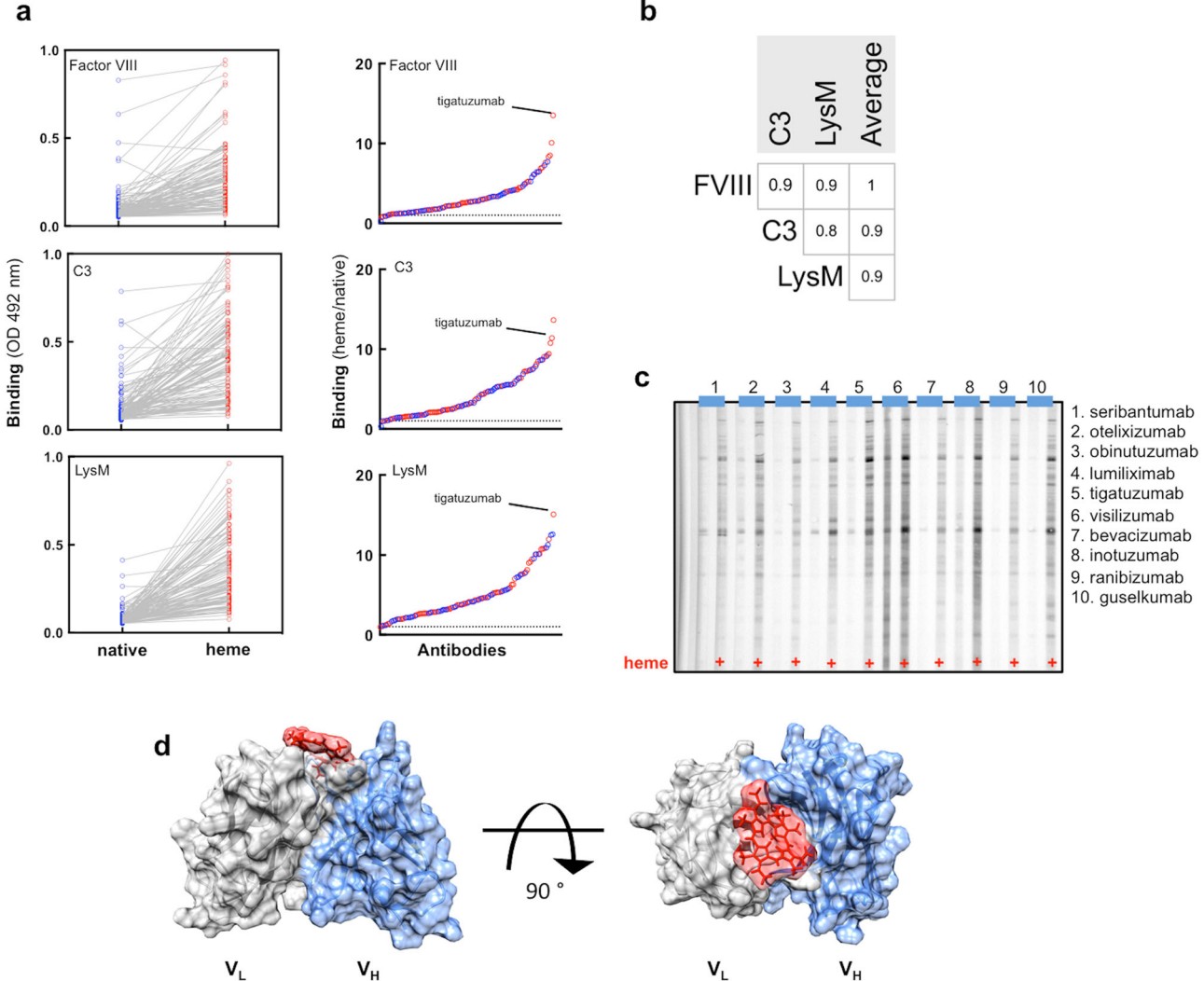

**Fig. 3 Heme-induced antigen-binding polyreactivity of Abs. a** ELISA analysis of the binding of the repertoire of therapeutic Abs to immobilized human (FVIII, C3) and bacterial (LysM AtlA) proteins. Each circle depicts the average reactivity of an individual Ab obtained from duplicate samples. The right panels show the global reactivity profile of the repertoire of therapeutic Abs. The protein binding intensity was obtained by dividing the reactivity of given Ab after exposure to heme by the reactivity of the same Ab in its native form. The dashed line represents a twofold increase in the binding. The red circles represent Abs that are currently in clinical trials. The blue circles indicate the clinically approved Abs. **b** Correlation between reactivities of the therapeutic Abs to different protein antigens upon exposure to heme. Matrix depicts the correlation analyses between the reactivity of heme-treated Abs to different proteins and the average reactivity toward the three proteins. The correlation analysis was performed using Spearman's rank-order test. The values of the correlation coefficients (ρ) are shown in the matrix. All P values are <0.005. The right panel illustrates the statistical significance (P values). **c** Immunoreactivity of selected Abs (top 10 heme-sensitive Abs) toward antigens present in the total lysate of *B. anthracis*. **d** The three-dimensional structure of tigatuzumab was modeled by sequence-based variable region modeling algorithm RosettaAntibody implemented on ROSIE web server (http://rosie.rosettacommons.org/). The putative heme-binding site was then predicted by docking of the protoporphyrin IX molecule to the variable region using the SwissDock web service protein–ligand docking platform-based on the EADock DSS algorithm (http://www.swissdock.ch/). The figure shows the side and top view of the variable region of tigatuzumab. Gray: variable domain of light chain; blue: variable domain of heavy chain. Heme is shown in red.

hydrophobic matrixes was assessed by chromatography). All these features have been shown to be of immense importance for the successful entry of Ab in the clinical practice[32,35].

The panel of therapeutic Abs used in the present study overlaps with the repertoire of Abs that was used in the work of Jain (113 of the studied monoclonal Abs correspond to identical samples). This allowed us to comprehensively analyze relationships between the physicochemical properties of the Abs assessed by different analytical techniques with their capacity to bind to heme or acquire polyreactivity upon exposure to heme. Statistical analyses revealed that the potential of therapeutic Abs to recognize immobilized heme strongly correlated with several biophysical properties of Abs (Fig. 7). Thus, Abs that have the propensity to

bind to heme were characterized with significantly higher natural polyreactivity, as assessed by three independent polyreactivity assays (PSR, ELISA, and BVP) (Fig. 7). Moreover, the increased binding to immobilized heme positively correlated with the tendency of the therapeutic Abs for self-binding and binding to other IgG molecules as estimated by three different assays (ACSINS, CSI, and CIC) (Fig. 7). These analyses revealed that the heme-binding Abs are generally aggregating at lower concentrations of ammonium sulfate, i.e., they are characterized by elevated propensity for aggregation mediated by hydrophobic interactions (SGAC100 assay, Fig. 7). Interestingly, a stronger association of heme had also a significant negative correlation with expression titer of the therapeutic Abs (Fig. 7).

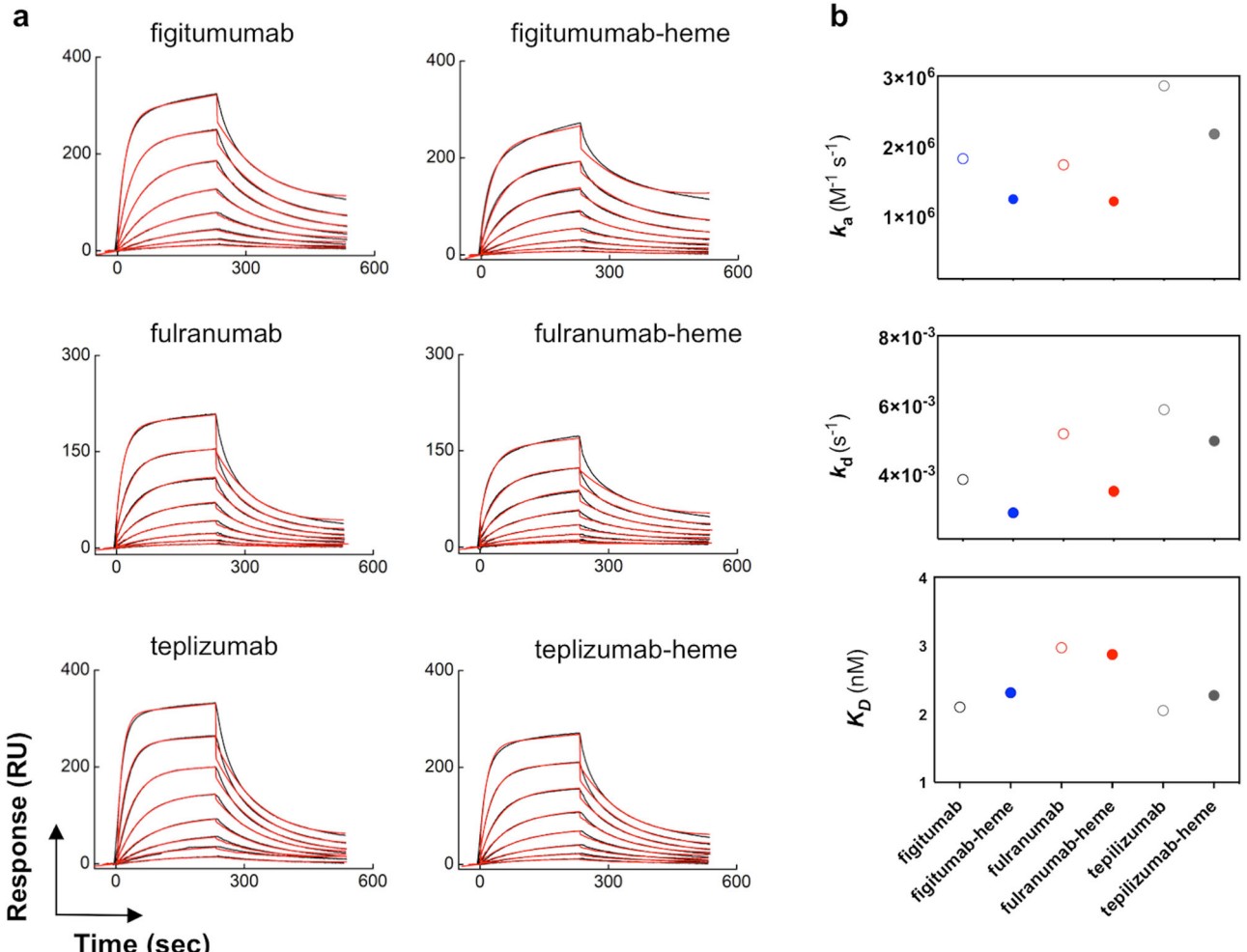

**Fig. 4 Interaction of heme-binding antibodies with human FcRn. a** Real-time binding profiles of selected heme-binding and heme-sensitive Abs to surface-immobilized human recombinant FcRn. The interaction analyses were performed with native and Abs exposed to an excess of hemin. The black lines depict the binding profiles obtained after injection of serial dilutions of monoclonal Abs, native, and after heme exposure (25–0.195 nM). The red lines depict the fits of data obtained by global analysis using the Langmuir kinetic model. All interaction analyses were performed at 25 °C. **b** Values of the kinetic rate constants of association (top panel), dissociation (middle panel), and equilibrium dissociation constant (bottom panel) for binding of native and heme-exposed monoclonal Abs to FcRn. The kinetics data were obtained after analyses of the real-time interaction profiles shown in (**a**).

Further, we investigated the relationship of biophysical properties of the Abs with heme-induced polyreactivity. The heme-sensitive Abs had significantly higher overall hydrophobicity, as determined by three independent assays (SGAC100, HIC, and SMAC). The Abs acquiring polyreactivity following contact with heme were also characterized by their intrinsic propensity for self-binding and cross-reactivity to human polyclonal IgG (Fig. 7).

Statistical analyses of results from different assays involving heme additionally suggested that heme-induced polyreactivity positively correlates with heme binding in solution and increase in ABTS oxidation (catalytic assay) (Fig. 7). Moreover, these analyses demonstrate that the study of Ab binding to immobilized heme and heme-induced Ab polyreactivity has considerably higher power in predicting developability issues of Abs as compared with the heme-binding assays in solution (absorbance spectroscopy and ABTS oxidation).

**Clinical status of the heme-binding Abs.** The studied panel of samples sourced V region sequences from therapeutic Abs molecules that have been already approved for clinical use or have undergone evaluation in phase II and phase III clinical trials. In previous work, it was shown that the Abs that entered clinical practice tend to be characterized by a significantly lower number of negative for therapeutic developability physicochemical and binding properties as compared with Abs that had merely reached clinical trials[32]. The threshold value of measures defining negative developability traits was set as the worst 10% of the value from different assays. By selecting Abs that bind heme with 10% highest intensity (11 Abs in total), we found that among these Abs only two molecules have been approved for clinical use (i.e., 18%, see Fig. 1b, Supplementary Table 1). Among the therapeutic Abs that bind less or do not bind at all to heme, 41% have been approved for use in clinic. Most of the Abs (9 out of 11) showing the highest heme-binding intensity crossed at least one of the negative thresholds set for other developability assays in the study by Jain et al.[32]. This result suggests that the interaction with heme is an indicator for the presence of liabilities of monoclonal Abs that might affect the final approval for clinical use.

**Discussion**
In the present study, we demonstrated that a considerable fraction of monoclonal therapeutic Abs recognizes heme. Following the exposure to heme, some of the monoclonal Abs acquired

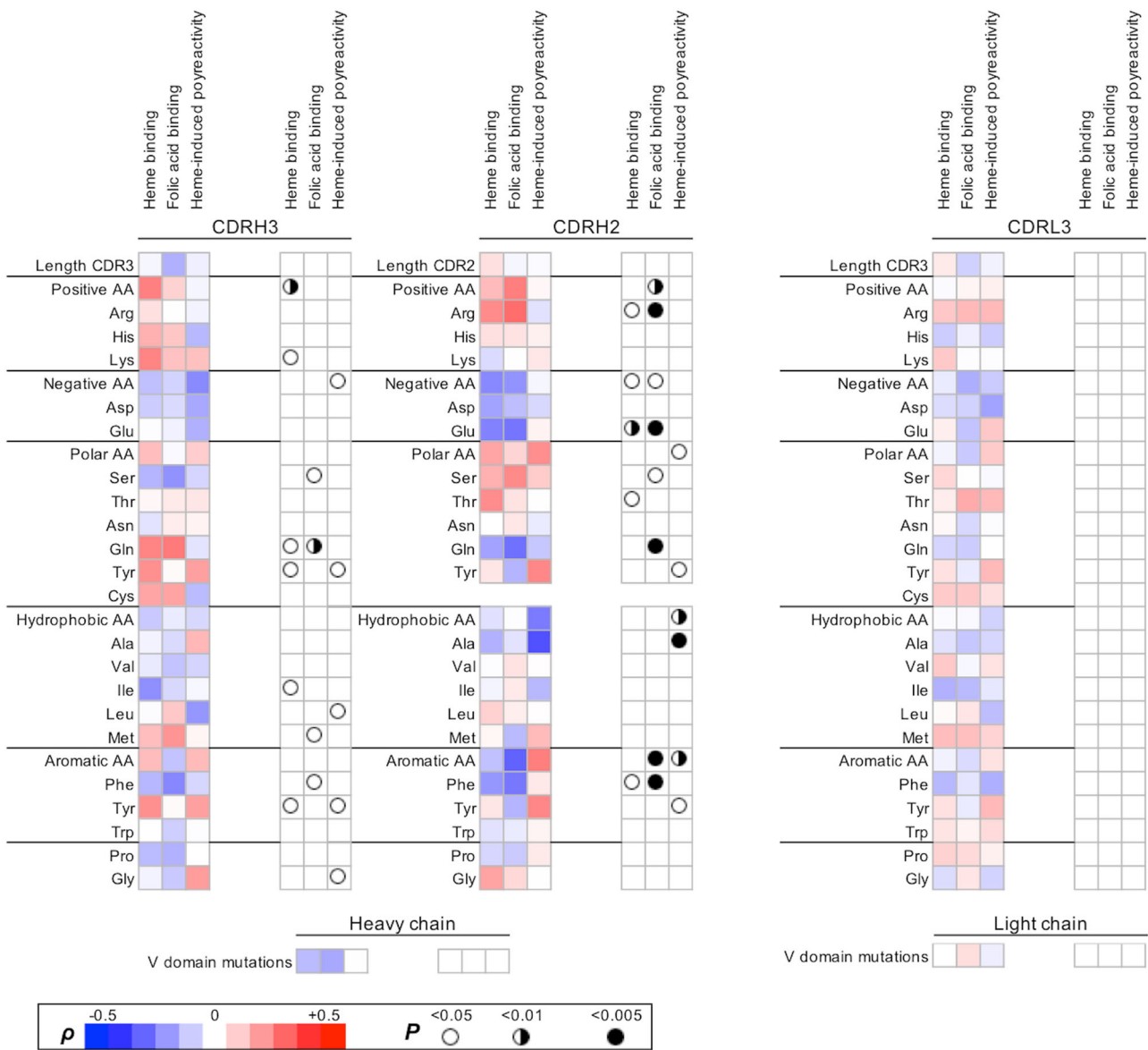

**Fig. 5 Correlation between interaction with heme and sequence features of the variable regions of therapeutic Abs.** The binding intensity to immobilized cofactors (heme and FA, Fig. 1b) and the degree of heme-induced antigen polyreactivity (Fig. 3a) were correlated with the sequence characteristics of heavy and light immunoglobulin chain variable regions of the studied therapeutic Abs. The heat map shows the values of the correlation coefficient ($\rho$) obtained after Spearman's rank-order analysis of the correlation of data from the binding assays with the number of amino acid replacements in heavy chain variable region; the lengths of CDRH3, CDRH2, and CDRL3; or the number of particular amino acid residues that are present in CDRH3, CDRH2, and CDRL3. The red color in the heat map signifies the positive value of the correlation coefficient, the blue-negative. The summary of statistical significance (P values) is depicted on the right panels.

antigen-binding promiscuity. We identified sequence patterns of the antigen-binding site that are associated with interaction with heme. Moreover, the interaction with heme significantly correlated ($P < 0.005$) with three different features of the Ab molecule, i.e., hydrophobicity, propensity for self-association, and intrinsic antigen-bidding polyreactivity. Although with lower significance ($P < 0.05$), heme binding also negatively correlated with expression titer of Abs in eukaryotic cells, a parameter strongly dependent on protein stability[32].

Previous studies have demonstrated that heme-binding sites on distinct proteins are enriched with specific amino acid residues[29,31,36]. Generally, heme has a tendency to associate with more hydrophobic regions of proteins[29]. Elucidation of sequence characteristics of the variable region of Abs that bind to heme revealed a positive correlation of the recognition of heme with the

number of Tyr residues in the most diverse region of the Ig molecule, i.e., CDRH3. The number of Tyr was also significantly elevated in the CDRH3 and CDRH2 of Abs that acquire polyreactivity after interaction with heme. Tyr is a well-known amino acid that can interact with heme. It is one of the amino acid residues particularly enriched in the heme-binding sites on proteins[29,30,37]. Tyr can interact with heme both by aromatic (π-stacking) interactions or by coordination of central iron ion (through its hydroxyl group). Aromatic amino acid residues such as tyrosine, phenylalanine, and tryptophan are usually localized in the interior of proteins. But Igs are unusual in that they carry a large number of aromatic amino acids exposed to the protein surface, and more specifically in CDRs[38,39]. Thus, the presence of Tyr in the antigen-binding site can provide appropriate environment for the binding of aromatic compounds such as heme

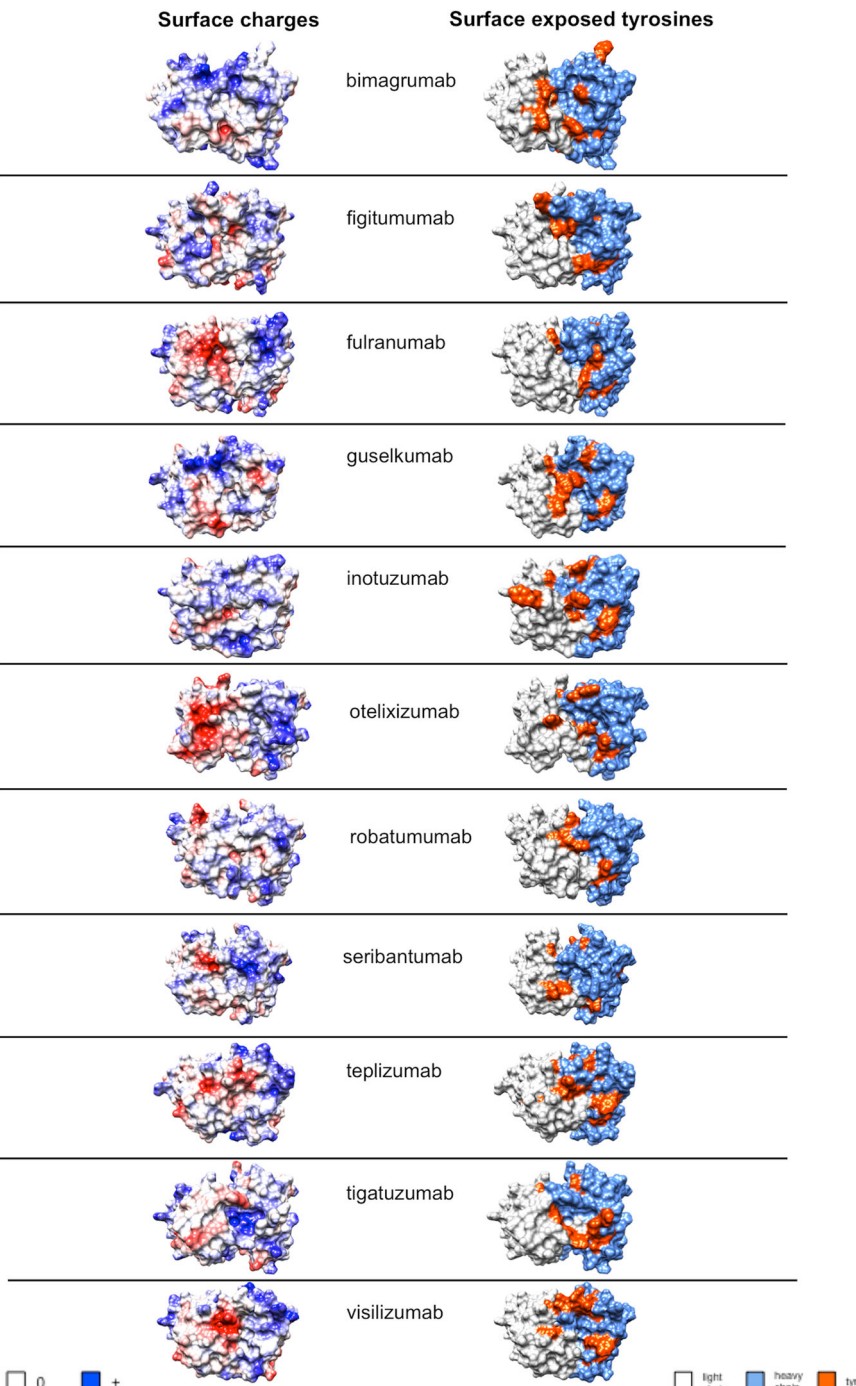

**Fig. 6 Molecular features of antigen-binding sites of selected heme binding and heme sensitive Abs.** (Left column) Coulomb's electrostatic potential of antigen-binding sites of selected Abs. The distribution of the electrostatic charges (blue for positive charge and red for a negative charge) was colored using UCSF Chimera software. The selected polyreactive Abs demonstrated the most pronounced binding to immobilized heme and/or heme-induced polyreactivity. (Right column) Structural models of antigen-binding sites depicting the surface exposed Tyr residues. Gray: variable domain of light chain; blue: variable domain of heavy chain, orange: tyrosine residues. The structural models were visualized by using UCSF Chimera software. All structural models of the variable immunoglobulin regions were built by ROSSIE online server.

(and other cofactors). Besides Tyr residues, the binding of Abs correlated with an elevated number of positively charged residues such as Lys and Arg in the CDRs. These residues can interact through ion bonds with carboxyl groups of heme (Fig. 1a), thus further facilitating the attachment of the heme molecule to IgG. It is noteworthy that conventional heme-binding sites on diverse proteins are also enriched in basic amino acid residues[29]. The sequence analyses also showed that Abs that bind heme and

acquire polyreactivity were depleted by amino acid residues with acidic side chains. This can be explained by the potential repulsion of these side chains with the carboxyl groups of heme. The lower number of acidic residues in CDR regions can also explain elevated hydrophobicity of the Abs.

Our data revealed that heme could specifically detect an elevated number of surface-exposed positive amino acid residues and aromatic residues in Abs. The capacity of heme to detect both

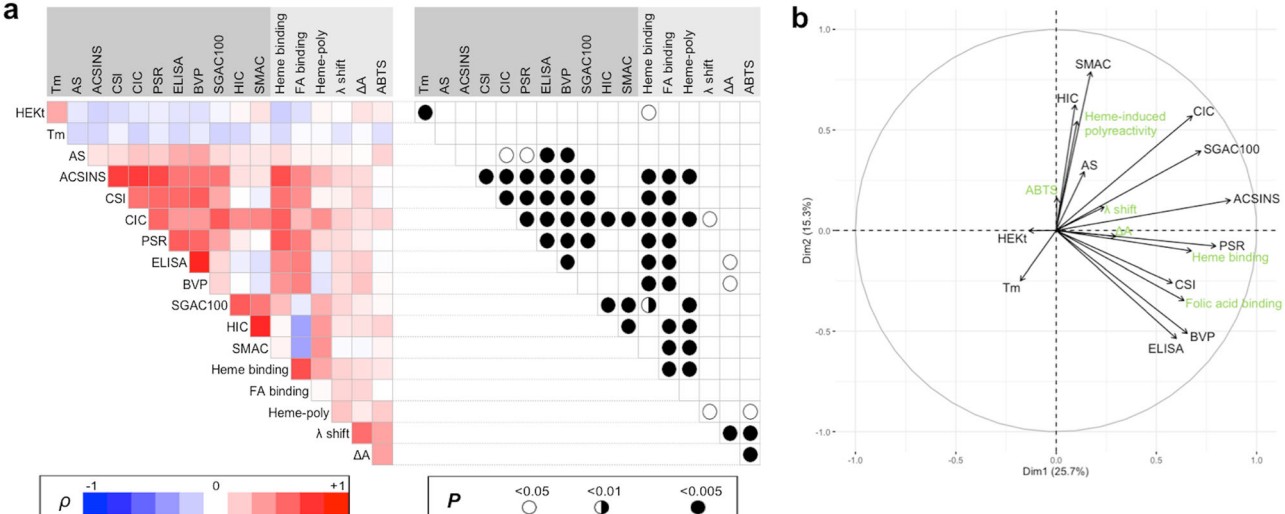

**Fig. 7 Analysis of the correlations among biophysical developability measurables and cofactors interactions parameters. a** Matrix depicting the correlation of 12 different Ab developability parameters with Ab-binding intensity to immobilized cofactors (heme and FA—binding), the degree of heme-induced antigen polyreactivity (heme-poly), and Ab interaction with heme in solution (λ shift, ΔA and ABTS). The heat map shows the values of correlation coefficient (ρ) obtained after Spearman's rank-order analysis of the correlation of data. The red color in the heat map signifies positive value of correlation coefficient, the blue—negative. The summary of statistical significance (P values) is depicted on the right panel. Data related to the biophysical developability measurables of the antibodies were obtained from the study of Jain et al.[32]. **b** Graphical output of the Principal Component Analysis where biophysical developability measurables (in black) and cofactors interactions parameters (in green) are represented as vectors in a plane defined by dimension 1 (horizontal axis) and dimension 2 (vertical axis), explaining 25.7% and 15.3% of the variability of the dataset, respectively. Projections of the vectors on each axis account for the quality of representation of the parameters by dimensions 1 and 2. Positively correlated parameters are grouped; negatively correlated parameters are represented in opposite quadrants and non-correlated parameters are shown by orthogonal vectors.

an increased presence of aromatic and positively charged amino acid residues in the V region well explains the strong correlation of certain physicochemical and functional parameters of V regions with the interaction with heme. Thus, the higher frequency of Tyr in the V regions was associated with an augmented hydrophobicity of the Abs and an increased tendency for homophilic binding (self-association)[40,41]. On the other hand, the higher prevalence of basic amino acid residues or surface patches with positive charge in the antigen-binding site has been demonstrated to correlate both with an antigen-binding polyreactivity and tendency for self-association[41–45]. These observations imply that heme preferentially interacts with V regions of Abs with particular molecular characteristics. Therefore, these results suggest that the molecular features of heme allow this compound to predict some important physicochemical and functional qualities of Abs. The utility of heme as a probe most probably stems from its distinctive chemical structure (Fig. 1a), allowing establishment of various types of non-covalent molecular interactions with proteins.

The present study also reveled that the set of clinical-stage Abs contains a fraction of molecules that specifically bind another cofactor—folate. Similarly as heme, the binding to folate correlated with some features of Abs such as a propensity for self-binding and polyreactivity. Notably, the reactivity to heme strongly correlated with reactivity to folate ($\rho = 0.69$, $P < 0.0001$). This suggests that the recognition of different heterocyclic compounds might be an intrinsic property of a specific fraction of Abs in Ig repertoires. Nevertheless, the Abs recognizing folate was a significantly lower number and this molecule was not able to predict the hydrophobicity of V region or expression titer of Abs, suggesting that heme has a superior capacity to characterize Abs.

Use of monoclonal therapeutic antibodies (Abs) for treatment of human diseases is an important milestone of modern medicine. Presently, approximately 90 therapeutic Abs are applied for the treatment of cancer, autoimmune, inflammatory, and infectious

diseases[46]. Hundreds of other Ab molecules are in clinical trials or are under development. The principal mechanism of therapeutic Abs is based on a high-fidelity recognition of a specific disease-associated target. To become a successful therapeutic, however, mere identification of an Ab molecule with a high binding affinity to a given target of interest is often not sufficient. Indeed, high-affinity recognition of antigen can be accompanied by various additional features of the Ab molecule that can have a negative impact on the production process, pharmacokinetics, and ultimate therapeutic efficacy. These features include expression yield, thermodynamic stability, propensity for aggregation, sensitivity to chemical degradation, immunogenicity, antigen-binding polyreactivity, and self-binding capacity[35,47–49]. The process of development of therapeutic monoclonal Abs is sophisticated, resource- and time-consuming[50]. Therefore, the use of simple predictive tools that can detect at early stage different developability issues is of high importance. Many approaches have been developed for in vitro or in silico evaluation of the various physicochemical features of Abs that can allow recognition of those with developability problems[51–59]. From these studies it is evident that only the simultaneous use of complementary analytical techniques would allow comprehensive interrogation of all possible developability issues of a particular Ab. In the study of Jain et al.[32], comprehensive analyses of 137 therapeutic Abs with an array of 12 complementary assays demonstrated the advantage of an integrative approach for successful identification of Abs with unfavorable biophysical and functional characteristics and therefore with low chances to enter clinical practice.

The results obtained in the present study provide evidence about ability of heme to detect simultaneously four critical liabilities for therapeutic Abs—antigen-binding polyreactivity, tendency for self-binding, hydrophobicity, and expression titer. Presence of any of these issues would compromise use of candidate therapeutic Ab in clinical practice. Thus, for example, elevated hydrophobicity and the tendency for self-association

would lead to increase aggregation of Abs, especially at high concentrations; polyreactivity would negatively impact pharmacokinetics and pharmacodynamics of Abs[35,52,60]. Since the heme-binding assay used in the present study is simple and allows high-throughput screening in a short time, large number of Abs can be assessed, thus saving time and resources associated with the employment of several complex analytical assays. The feasibility of binding to heme as an analytical tool for detection of Ab liabilities was also suggested by the fact that currently only 18% of top heme-binding Abs has been approved for clinical practice, whereas 41% of Abs were approved among therapeutic Abs with lower or no heme binding capacity.

Our study also emphasized that some therapeutic Abs can change their stringent specificity upon contact with heme. This observation extends the findings from the study of McIntyre and Faulk[61] whereby analyses of a set of nine therapeutic Abs approved for the treatment of cancer, it was demonstrated that most of the Abs acquired substantial reactivity to different autoantigens after contact with heme. The appearance of polyreactivity upon exposure to heme can have repercussions for therapeutic Abs—it can decrease potency and represents a risk for side effects due to compromised pharmacokinetics and off-target binding[52,60]. Therefore the assessment of heme-induced polyreactivity provides another dimension for analyses of Ab developability. This cryptic therapeutic Ab liability can be of relevance in multiple disease conditions associated with hemolysis and tissue damage, where extracellular heme is available, thus compromising the efficacy of the therapeutic Ab with an inherent potential to acquire polyreactivity. Of note, high concentrations of extracellular heme, i.e., 20–50 μM were reported in patients with certain hemolytic conditions[25,62]. These concentrations are well beyond those used in the present study for induction of polyreactivity of Abs.

In conclusion, the present study demonstrated that a significant fraction of clinical-stage Abs interacts with heme. Specific sequence traits identify the set of cofactor-binding Abs. Moreover, those Abs that were able to interact with heme presented with particular physicochemical and functional qualities. This study contributes to a better understanding of the fraction of Abs that recognizes cofactors. It may also have repercussions for applied science as heme molecule has the capacity to predict several liabilities of monoclonal Ab that can compromise their introduction into the clinical practice.

## Methods

**Antibodies**. In the present study, we used 113 samples with variable region sequences corresponding to therapeutic antibodies that are approved for use in clinic or preceded as far as Phase II and Phase III clinical trials, with vesencumab being the lone exception. All antibodies were expressed as human IgG1 in HEK293 cells. A detailed description of production of this set of recombinant antibodies was provided in Jain et al.[32]. All reagents used in the study were with the highest purity available.

**Antibody reactivity to surface-immobilized heme**. For covalent in situ immobilization of heme or folic acid through their carboxyl groups, we applied the amino-coupling procedure described in ref. [63]. In brief, 96-well polystyrene plates NUNC MaxiSorp™ (Thermo Fisher Scientific, Waltham, MA) were incubated for 1 h at RT with 100 μl/well of the solution in PBS of 0.5% laboratory-grade gelatin (J.T. Baker Chemicals, Fisher Scientific). After 3 washes with deionized water, 50 μl/well of 1 mM solution of oxidized for heme, i.e., hemin (Fe(III)-protoporphyrin IX chloride, Frontier Scientific, Inc., Logan, UT) or alternatively—folic acid (FA, Sigma-Aldrich, St. Louis, MO) were added. Both substances were diluted in a 1:1 DMSO:water mixture. To wells containing hemin, FA or vehicle only, 25 μl/well of 60 mM water solution of 1-ethyl-3-(3-dimethylaminopropyl) carbodiimide hydrochloride (EDC, Thermo Fisher Scientific) were added, resulting in final concentration of 20 mM. The conjugation reaction between activated carboxyl groups of heterocyclic compounds and amino groups from gelatin was left to occur for 2 h at RT, in dark, with periodic gentle shaking of the plates. After, the plates were washed three times with deionized water. To inactivate EDC-activated carboxyl groups on gelatin, all wells were incubated for 5 min with 100 μl/well of

excess (1 M solution) of ethanolamine, pH 8, and then washed thoroughly with deionized water. Next, plates were incubated for 1 h with PBS containing 0.25% Tween 20. All analyzed recombinant IgG1 antibodies were diluted to 20 μg/ml in PBS containing 0.05% Tween 20 (PBS-T) and incubated for 1 h at RT with surface immobilized heme (or FA). As a control, each antibody was also incubated with wells coated with gelatin only. The plates contained also a standard, which represented 100 μg/ml of pooled human IgG (IVIg, Endobulin, Baxter, USA) incubated with surface immobilized heme (or FA). In another experimental setting, selected antibodies demonstrating the highest heme-binding activity were analyses at serial (3× each step) dilutions starting from concentration of 30 μg/ml (200 nM). After incubation with antibodies, the plates were washed at least 5 times with PBS-T and then incubated for 1 h at RT with HRP-conjugated anti-human IgG (Fc-specific, clone JDC-10, Southern Biotech, Birmingham, AL) diluted 3000× in PBS-T. Following washing with PBS-T, the immunoreactivity of the antibodies was revealed by the addition of peroxidase substrate, o-phenylenediamine dihydrochloride (Sigma-Aldrich). The measurement of the optical density at 492 nm was performed after the addition of 2 M HCl with a microplate reader (Infinite 200 Pro, Tecan, Männedorf, Switzerland).

**Real-time analyses of the interaction of heme with antibodies**. SPR-based assay (Biacore 2000, Biacore GE Healthcare, Uppsala, Sweden) was applied to elucidate kinetics of interaction of heme with therapeutic antibodies. Antibodies demonstrating substantial binding to immobilized heme—bavituximab, bimagrumab, brodalumab, figitumumab, fulranumab, guselkumab, robatumumab, teplizumab, tigatuzumab, and visilizumab were covalently coupled to surface of CM5 sensor chips (Biacore) using an amino-coupling kit (Biacore). In brief, the antibodies were diluted in 5 mM maleic acid solution, pH 4 to a final concentration of 20 μg/ml, and injected for 7 min over sensor surfaces pre-activated by a mixture of EDC/N-hydroxysuccinimide. Unconjugated carboxyl groups were saturated by injection of 1 M solution of ethanolamine.HCl. All measurements were performed using HBS-EP buffer with following composition: 10 mM HEPES pH 7.2; 150 mM NaCl; 3 mM EDTA, and 0.005 % Tween 20. The binding analyses were performed at temperature of 25 °C. Hemin (1 mM stock solution in 0.05 NaOH) was always sequentially diluted in HBS-EP to concentrations of 1250, 625, 312.5, 156.25, 78.1, 39, and 19.5 nM. For each concentration, the dilution sequence was repeated immediately before injection. The buffer flow rate during all real-time interaction measurements was set at 30 μl/min. The association and dissociation phases of the binding of hemin to the immobilized antibodies were monitored for 5 min. The binding of hemin to reference channel containing activated and deactivated carboxymethylated dextran only was used as negative control and was subtracted from the binding during data processing. The sensor chip surfaces were regenerated by 30 s exposure to a solution 3 M solution of KSCN (Sigma-Aldrich). The assessment of binding kinetics was performed by BIAevaluation version 4.1.1 Software (Biacore) by using Langmuir binding model.

**Analysis of polyreactivity of therapeutic antibodies following exposure to heme**

*Enzyme-linked immunosorbent assay*. Ninety-six-well polystyrene plates NUNC MaxiSorp™ (Thermo Fisher Scientific) were coated for one hour at RT with human recombinant factor VIII (Advate, Baxter), human plasma-derived C3 (CompTech, Tyler, TX) and recombinant LysM polypeptide of AtlA from *Enterococcus faecalis* (kindly provided by Dr. Stephan Mesnage, University of Sheffield, UK). All proteins were diluted to 3 μg/ml in PBS. After blocking by incubation for one hour with 0.25 % solution of Tween 20 in PBS, the plates were incubated with therapeutic antibodies. Each antibody was first diluted in PBS to 100 μg/ml, exposed or not for 10 min to 5 μM of hemin (stock solution in DMSO), and then diluted to 20 μg/ml in PBS-T and incubated with coated proteins for 90 min at RT. As an internal control each plate was incubated with 250 μg/ml of native pooled human IgG. For assessment of binding of each recombinant antibody in native and heme-exposed form to the studied proteins, the plates were first washed 5 times with PBS-T and then incubated for 1 h at RT with HRP-conjugated anti-human IgG (Fc-specific, clone JDC-10, Southern Biotech, Birmingham, AL) diluted 3000× in PBS-T. Following washing with PBS-T, the immunoreactivity of the antibodies was revealed by addition of peroxidase substrate, o-phenylenediamine dihydrochloride (Sigma-Aldrich). The measurement of the optical density at 492 nm was performed after addition of 2 M HCl with a microplate reader (Infinite 200 Pro, Tecan, Männedorf, Switzerland).

*Immunoblot*. Lysate of *Bacillus anthracis* was loaded on a 4–12% gradient NuPAGE Novex SDS-PAGE gel (Invitrogen, Thermo Fisher). After migration, proteins were transferred on nitrocellulose membranes (iBlot gel transfer stacks, Invitrogen, Thermo Fisher) by using iBlot electrotransfer system (Invitrogen, Thermo Fisher). Membranes were blocked overnight at 4 °C in PBS buffer containing Tween 0.1% (PBS-T). Next, the membranes were fixed on Miniblot system (Immunetics, Cambridge, MA) and incubated for 2 h at RT with 10 μg/ml of selected native or heme-exposed therapeutic antibodies. The antibodies were pre-treated at 50 μg/ml in PBS with 5 μM hemin for 10 min and diluted in PBS-T before loading. Membranes were washed (6 × 10 min) with excess of PBS-T before being incubated for 1 h at RT with an alkaline phosphatase-conjugated goat anti-human IgG (Southern Biotech, Birmingham, AL), diluted 3000× in PBS-T. Membranes were then

thoroughly washed (6 × 10 min) with PBS-T before revealed with SigmaFast NBT/BCIP substrate solution (Sigma-Aldrich).

**Real-time analyses of interaction of antibodies with human FcRn.** The binding of selected heme-sensitive antibodies to human FcRn before and after exposure to heme was studied by SPR-based biosensor technology (Biacore). Biotinylated recombinant human FcRn (Amsbio LLC, Cambridge, MA) was diluted to 5 μg/ml in running buffer—100 mM Tris pH 6 (adjusted by addition of dry citric acid), 100 mM NaCl, 5 % Glycerol, and 0.1% Tween 20—and injected with a flow rate of 10 μl/min over the pre-activated sensor chip surface, coated with streptavidin by the manufacturer (SA chip, Biacore). We aimed to achieve approximately 0.5 ng/mm$^2$ immobilization density on the sensor surface. For kinetics analyses, the therapeutic antibodies figitumumab, fulranumab and teplizumab were first diluted in PBS to 100 μg/ml (670 nM) and treated with 6.7 μM of hemin (1 mM stock solution in DMSO) or with vehicle only. Following 5 min incubation on ice, the native and heme-exposed antibodies were serially diluted in the binding buffer to the following concentrations—25, 12.5, 6.25, 3.125, 1.56, 0.78, 0.39, and 0.195 nM. The interaction with surface immobilized FcRn was performed at a flow rate of 30 μl/min. The association and dissociation phases of the binding of antibodies at different concentrations to the immobilized FcRn were monitored for 4 and 5 min, respectively. For regeneration of the binding surface, the chip was exposed for 30 s to a solution of 100 mM TRIS pH 7.7/100 mM NaCl and this step was repeated twice for the highest concentration of samples (25 and 12.5 nM). The analysis was performed at 25 ℃. The assessment of binding kinetics was done by BIAevaluation version 4.1.1 Software (Biacore) by using global kinetics analyses and Langmuir binding model.

**Absorbance spectroscopy.** We used high-throughput absorbance spectroscopy to elucidate the interaction of the repertoire of therapeutic antibodies with hemin in solution. The recombinant antibodies were first diluted to 200 μg/ml in PBS. The dilutions were performed directly in UV-star 96-well microplates (Greiner Bio-One, Les Ulis, France). To 100 μl of antibody solution, an equal volume of freshly prepared hemin solution (10 μM in PBS) was added and intensively homogenized. The resulting concentration of antibodies was 100 μg/ml (670 nM) and of heme was 5 μM. As control heme was added to buffer only. The plates were incubated for 30 min at RT in dark. The absorbance spectra in the wavelength range between 350 and 450 nm were recorded by using Tecan Infinite 200 Pro, the microplate reader. The spectral resolution was 2 nm. All measurements were done at RT. To assess the changes in spectral properties of heme upon interaction with antibodies, the shift in maximal absorbance wavelength was defined as follows: λ-shift = $\lambda_{max}$ of hemin alone—$\lambda_{max}$ of hemin in the presence of antibody. The differential spectra (absorbance spectrum of hemin in the presence of antibody—absorbance spectrum of hemin alone) allowed quantification of the maximal increase in the absorbance intensity (ΔA).

**Peroxidase assay.** To evaluate how interaction with antibodies influences the intrinsic peroxidase activity of heme, we performed colorimetric catalytic assay. The antibodies from the studied repertoire were first diluted in PBS to 100 μg/ml (670 nM) and exposed to 5 μM final concentration of hemin. Following 90 min incubation at RT, 50 μl of antibody exposed to hemin or hemin in PBS only were mixed with 150 μl of reaction buffer (0.15 M citrate-phosphate buffer pH 5) containing 0.9 mM of 2,2-Azino-bis(3-ethylbenzothiazoline-6-sulfonic acid) diammonium salt (ABTS, Sigma-Aldrich) and 6 mM H$_2$O$_2$. The reaction was performed in 96-well microtiter plates at RT in dark. The absorbance at 414 nM was recorded at 10 and 20 min after mixing by using Tecan Infinite 200 Pro, the microplate reader. The peroxidase activity of bovine serum albumin at 150 μg/ml exposed to 5 μM hemin was used as a control.

**Structural modeling of the variable regions of antibodies and molecular docking.** For modeling of three-dimensional structures of the variable domains of heavy and light chains of selected heme-binding and heme-sensitive antibodies, we used sequence-based modeling algorithm—RosettaAntibody3 program[64–67]. The amino acid sequences of the variable regions were loaded to ROSIE online server (http://antibody.graylab.jhu.edu/). Relaxed 3D models with the lowest free energy were visualized by using the Chimera UCSF Chimera package. Chimera is developed by the Resource for Biocomputing, Visualization, and Informatics at the University of California, San Francisco (supported by NIGMS P41-GM103311)[68]. Chimera software was also used for visualization of Coulombic electrostatic potential of antigen-binding sites as well as to highlight the surface exposed Tyr residues in the antigen-binding site.

The structural model of the antibody variable region of tigatuzumab was used as an input for docking of protoporphyrin IX. To this end, we used the SwissDock web service, which uses the CHARM force field and it is dedicated to the prediction of protein–small molecule interaction[69,70]. The three-dimensional structures of the most probable complex were visualized by using Chimera software.

**Statistical analyses and reproducibility.** Abs that meet the following criteria were included in statistical analyses: availability of sequence information (obtained from IMGT/mAb-DB database (http://www.imgt.org/mAb-DB/) or from the article of

Jain et al.[32], availability of the interaction parameters obtained in the present study, and availability of data about biophysical measurements performed in the study of Jain et al.[32]. Estimation of the composition and length of CDR regions, as well as number of amino acid replacements in V domains, was performed by IMGT/JunctionAnalysis tool[71,72].

The antibody binding intensity to immobilized heme and FA, average degree of antibody binding to protein antigens (FVIII, C3, and LysM) after heme exposure, increase in absorbance intensity and the wavelength number shift in maximal absorbance at Soret band as well as fold increase in the catalytic activity of complexes of heme with antibodies as compared to heme also were used as input for correlation analyses (source data are presented in Supplementary Data 2). Thus experimental measurable of therapeutic Abs were correlated with different features of variable regions—number of somatic mutations in $V_H$ and $V_L$ domains, lengths of CDRH3, CDRH2, and CDRL3, number of charged, polar, aromatic, and hydrophobic residues in the CDR regions, and frequency of individual amino acid residues in CDR regions. Correlation analyses were performed by nonparametric Spearman's rank-order analysis using Graph Pad Prism v.6 software (La Jolla, CA). A significant correlation was considered only if the $P$ value ≤ 0.05. Principal component analysis was performed on R software (http://www.R-project.org/) with the FactoMineR package[73]. As per Jain et al.[32], values for SGAC100 were negated before applying PCA.

**Reporting summary.** Further information on research design is available in the Nature Research Reporting Summary linked to this article.

## Data availability
The data that support the findings of this study are available from the corresponding author upon reasonable request.

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

## Acknowledgements

All the antibodies characterized in this work were produced, purified, and subjected to quality control through the combined efforts from the molecular core, high throughput expression, and protein analytics departments at Adimab LLC. We are grateful to Dr. Max Vásquez, Dr. Yingda Xu, and Dr. Tushar Jain (Adimab, USA) for their vital support and discussions. This work was supported by Institut National de la Santé et de la Recherche Médicale (INSERM, France), Centre National de la Recherche Scientifique

(CNRS, France), by the European Research Council (Project CoBABATI ERC-StG-678905 to J.D.D.), and by a grant from "Fondation ARC pour la recherche sur le cancer", France (PJA 20171206410).

## Author contributions

J.D.D. conceived the project, designed the experiments, performed experiments, and data analysis, and wrote the paper. M.L. designed the experiments, performed experiments, and data analysis, and wrote the paper. A.K. and S.R. performed the experiments and the data analysis.

## Competing interests

The authors declare no competing interests.
