## [Peer Review File · Communications Biology]

Reviewers' comments:

Reviewer #1 (Remarks to the Author):

In this manuscript, Lecerf et al. analyze 113 antibodies of therapeutic interest (in clinical or pre-clinical stages) and analyze their binding to the heme group (and to folic acid). The rationale for this is that antibodies which show crossreactivity to heme are more prone to be problematic when developed clinically. The authors show that a big percentage of the antibodies which apparently turned out to be problematic had strong reactivity to heme and they point out that this could be an easy way to predict developability of an antibody. I found the study sound, well-designed and of interest. The data is clear and convincing, and the manuscript well written. I found important that the authors showed that the interaction with heme occurred both when immobilized on a plate and when in suspension (since the binding/coupling could have made new epitopes responsible for the binding). Interestingly the heme group seems to interact with the variable domain of the antibodies and after binding most of the antibodies showed increased crossreactivity.

I only have minor comments and some questions:

- While I find the manuscript of interest and see the value of using the heme reactivity as a predictor for developability, how would the authors envision that? Even if an antibody binds heme it may still be very helpful in the clinic. In this regard, and merely out of curiosity, is there any free heme naturally released to the blood (under normal non-diseased conditions) that could engage the undesired binding or crossreactivity the authors show in their study after intravenous inoculation of a therapeutic antibody?

- The authors used a repertoire of 113 monoclonal therapeutic antibodies expressed as human IgG1. Were they all of human origin? At least those mentioned later on in the manuscript with the suffix -umab should be of human origin. In this regard, I think a list of antibody names and mentioning a few characteristics, like use and so on, would be helpful to the reader. Perhaps also show the CDRs or variable domains of at least the most problematic antibodies. This could be added as supplementary material.

- Were all the Fc regions identical in the 113 antibodies? Or were they bearing the original one from each corresponding antibody?

- Figure 6: I think it would be nice to have a legend and some column titles added.

- Page 7: Authors picked some proteins to test crossreactivity. Was there any specific rationale for those that were picked? Would, under the authors' opinion, some binding to proteins from *Enterococcus faecalis* be expected since the bacterium may reside in human intestines?

- Page 9: I found interesting that no statistically significant correlation was observed between frequency of any amino acid residue type in CDRL3 and the binding to heme or folate. Did the docking/structural data show where the CDRL3 was positioned?

- Page 9: I feel that the word "respectively" should be added at the end of the following sentence to better reflect the manuscript's results: "...lower frequencies of Ala and Leu in CDRH2 and CDRH3, respectively".

- The binding of native and heme-exposed Abs to hFcRn showed no significant differences in the binding affinity (Fig. 4). The data is clear, no differences in the KD, but there were some in the Ka and Kd. What could be an interpretation for that if the heme binds the variable domains?

- Figure 5 and Page 8: When a figure reads "aa Lys", would that be the number of Lys present in that variable domain or just whether is present at all?

- Figure 3B: Perhaps there's no need for a color scale in this panel, if all the values shown are deep red (couldn't really tell in my screen). Just indicate the values. Same for the p-values since they are all below 0.005. I'll leave it up to the authors since I understand the legend keeps the

aesthetics and uniformity of the figures.

- I found some petty typos:

Page 3: "explained" should read "explain".

Page 4: "parts of heme molecule" should read "parts of the heme molecule"

Page 8: "for the numbers of" should read "for the number of"

Page 10: "measured features of Abs are include" should have either "are" or "include" removed.

Page 14: "rcontains" should read "contains".

Page 16: "et al" should be followed by a period.

Page 17: "demonstrated that significant fraction" should read "demonstrated that a significant fraction".

Page 20: "Adrich" should read "Aldrich".

Page 32: "righ" should read "right".

Page 32: "Graph Pad" should read "GraphPad".

Reviewer #2 (Remarks to the Author):

In their manuscript, Lecerf et al. show that a fraction of monoclonal therapeutic antibodies recognizes heme and that following the exposure to heme some of the therapeutic antibodies acquire antigen-binding promiscuity. In particular, they identify the amino acids in the antibody sequences most responsible for the interaction with heme. Considering the growing importance and interest in antibodies as therapeutics, and the fact that the development of therapeutic antibodies is challenging and time-consuming, being able to have strategies that quickly and effectively can predict early-stage different developability issues is of extreme interest. The authors state that the non-specific binding of antibodies to heme can be used as a replacement for four commonly used assays to verify the developability of therapeutic candidates. Moreover, they address how the increased non-specificity of some therapeutic antibodies, when exposed to heme, can represent a risk for the use of such molecules, especially for the treatment of diseases where hemolytic conditions are present.

I do think that in the antibody engineering and development field having assays that can identify developability issues in very early phases is critical, not sure if the heme assay is ready to replace some of them or it's one that might be considered to be added.

I found interesting the increase of polyreactivity of some antibodies in the presence of heme, which I agree with the authors might be a critical issue for therapy of diseases where hemolysis might be present, although I don't think this aspect of the manuscript has been deeply explored. It would have been interesting to see how and if this polyreactivity plays a role in an in vivo or ex vivo model.

Comments

Figure 1B: a) it would be interesting to show on the graphs the 44 antibodies among the 113 tested that are currently in clinical use, to clearly see if therapeutic antibodies behave better in a poly-specificity assay; b) in the results section (page 5 line 106) heme-binding antibodies were considered the one with a 10-fold higher intensity to immobilized heme as compared to the control protein alone. Why was the 10-fold chosen? Why in the figure a dashed line represents the 2-fold increased binding and not the 10-fold?

Figure 1D: in the text (page 6 line 111) it's reported that the antibodies recognizing heme with the highest intensity were subjected to additional analysis and the binding profile of 10 antibodies is shown in figure 1C. However, in figure 1D the sensorgrams of only fulranumab and figitumumab are shown, although in the figure legend tigatuzumab is also mentioned. I think it would be interesting to see the sensorgrams of all the 10 antibodies, or at least a table reporting the affinities measured for all 10 antibodies.

Figure 2: similar to figure 1, it would be interesting to show on the graphs the 44 antibodies among the 113 tested that are currently in clinical use.

Figure 3: similar to figure 1, it would be interesting to show on the graphs the 44 antibodies among the 113 tested that are currently in clinical use.

Although most of the antibodies have a binding signal below 0.5 absorbances before and after the exposure to heme, most of them do seem to have an increase in binding. For two of the tested targets, there are a few antibodies whose polyreactivity seems to be reduced after exposure with heme. Any explanation of those behaviors?

Is there a correlation between non-specific binding to heme and the acquisition of antigen-binding polyreactivity? What is the percentage of the total antibodies that will be identified combining the two tests? Can the heme-induce-polyreactivity be correlated with "classical" developability assays?

Figure 4: the affinities for the most heme-reactive antibodies is between ~300-500 nM, why for this assay heme was used at 25-0.195 nM?

Minor comments

Page 5 line 97: Heme is referred to as "Fe-protoporphyrin IX", while in figure 1 legend as "Fe(II)protoporphyrin IX" and page 18 line 387 as "Fe(III)-protoporphyrin IX". It would be better to have consistency in the nomenclature and to clarify if it is Fe(II) or Fe(III).

Page 14 line 308: there is a typo "Abs rcontains".

Reviewers' comments:

Reviewer #1 (Remarks to the Author):

In this manuscript, Lecerf et al. analyze 113 antibodies of therapeutic interest (in clinical or pre-clinical stages) and analyze their binding to the heme group (and to folic acid). The rationale for this is that antibodies which show crossreactivity to heme are more prone to be problematic when developed clinically. The authors show that a big percentage of the antibodies which apparently turned out to be problematic had strong reactivity to heme and they point out that this could be an easy way to predict developability of an antibody. I found the study sound, well-designed and of interest. The data is clear and convincing, and the manuscript well written. I found important that the authors showed that the interaction with heme occurred both when immobilized on a plate and when in suspension (since the binding/coupling could have made new epitopes responsible for the binding. Interestingly the heme group seems to interact with the variable domain of the antibodies and after binding most of the antibodies showed increased crossreactivity.

We are thankful to the Reviewer for these positive comments regarding our study.

I only have minor comments and some questions:

- While I find the manuscript of interest and see the value of using the heme reactivity as a predictor for developability, how would the authors envision that? Even if an antibody binds heme it may still be very helpful in the clinic. In this regard, and merely out of curiosity, is there any free heme naturally released to the blood (under normal non-diseased conditions) that could engage the undesired binding or crossreactivity the authors show in their study after intravenous inoculation of a therapeutic antibody?

Under physiological conditions the erythrocytes are disposed in the spleen without any release of free hemoglobin or heme. In cases of accidental hemolysis in plasma there are high affinity scavengers of hemoglobin and heme i.e. haptoglobin and hemopexin, that would prevent availability of free heme. Thus, there is low probability that in absence of hemolysis antibodies will be exposed to heme.

However, under chronic intermittent intravascular hemolysis (as observed in severe hemolytic disorders) it was reported that heme and hemoglobin- scavenging proteins are depleted. In these conditions there is high probability that antibodies are exposed to free heme in vivo. Importantly, some cancers as chronic lymphocytic leukaemia that are treated with monoclonal antibodies can be accompanied by hemolytic disease (hemolytic anemia).

Moreover, one can speculate that as monoclonal antibodies are frequently given as therapy in combination with cytotoxic drugs (chemotherapy). The massive cell death in these can result in release of heme from dying cells and thus influence the antibodies.

- The authors used a repertoire of 113 monoclonal therapeutic antibodies expressed as human IgG1. Were they all of human origin? At least those mentioned later on in the manuscript with the suffix -umab should be of human origin. In this regard, I think a list of antibody names and mentioning a few characteristics, like use and so on, would be helpful to the reader. Perhaps also show the CDRs or variable domains of at least the most problematic antibodies. This could be added as supplementary material.

We are grateful to the Reviewer for this constructive remark. We prepared a supplementary table, including the sequence of variable regions of monoclonal antibodies, their target specificities, their origin (human, or humanized), sequences of CDR H3 and L3, their clinical status, their original isotype etc. This table also highlights the antibodies that demonstrated the most pronounced binding to heme. We present the table as a supplementary material along with the revised version of the manuscript.

- Were all the Fc regions identical in the 113 antibodies? Or were they bearing the original one from each corresponding antibody?

Regardless of their original IgG subclass, all therapeutic antibodies used in this study were expressed as IgG1. Thus, they vary only by the sequence of their variable region of heavy chain.

- Figure 6: I think it would be nice to have a legend and some column titles added.

We have edited Figure 6 and added a legend and titles of the columns.

- Page 7: Authors picked some proteins to test crossreactivity. Was there any specific rationale for those that were picked? Would, under the authors' opinion, some binding to proteins from *Enterococcus faecalis* be expected since the bacterium may reside in human intestines?

We selected the panel of antigens for evaluation of polyreactivity based on our previous experience with these antigens. The three proteins used in ELISA experiments (two human and one with bacterial origin) are structurally and sequence unrelated and hence they are good model for evaluation of polyreactivity of antibodies. We do not expect bias in reactivity towards *Enterococcus faecalis* – bacterium that resides in human intestines, because the panel of therapeutic antibodies are artificially created after immunization of animals or selection from combinatorial antibody libraries.

- Page 9: I found interesting that no statistically significant correlation was observed between frequency of any amino acid residue type in CDRL3 and the binding to heme or folate. Did the docking/structural data show where the CDRL3 was positioned?

The docking model predicts that heme binding occurs in the centre of the antigen-binding site. The interaction involves contacts both with heavy and light Ig chains. Based on our data, however, we cannot extrapolate and explain why the sequence of CDR L3 did not show significance in heme binding. One possible explanation is that this region has lower diversity and lower variation in the length as compared with CDR H3. We now added this interpretation in the revised version of the manuscript.

- Page 9: I feel that the word “respectively“ should be added at the end of the following sentence to better reflect the manuscript's results: “...lower frequencies of Ala and Leu in CDRH2 and CDRH3, respectively”.

As recommended by the Reviewer we added the word “respectively” to the pointed sentence.

- The binding of native and heme-exposed Abs to hFcRn showed no significant differences in the binding affinity (Fig. 4). The data is clear, no differences in the KD, but there were some in the Ka and Kd. What could be an interpretation for that if the heme binds the variable domains?

We performed a recent study, where the mechanism of interaction of human IgG1 with FcRn was elucidated (*J Immunol*, 2020, 205:2850). The obtained results demonstrated that formation of complexes of antibodies with their antigens could cause long distance effects that influence the FcRn binding site. Analogously as the present results with heme, we observed that complexes of antibodies with their protein antigens resulted in compensation of changes in the rate constants, further resulting in absence or only little changes in equilibrium affinity.

- Figure 5 and Page 8: When a figure reads “aa Lys”, would that be the number of Lys present in that variable domain or just whether is present at all?

This presents the correlation of the number of lysines in CDRs with the numerical values of heme binding, folate binding or heme-induced reactivity.

- Figure 3B: Perhaps there’s no need for a color scale in this panel, if all the values shown are deep red (couldn’t really tell in my screen). Just indicate the values. Same for the p-values since they are all below 0.005. I’ll leave it up to the authors since I understand the legend keeps the aesthetics and uniformity of the figures.

We agree with the Reviewer that colour legend and signs for the p-values are not necessary for this particular figure. In the revised version of the manuscript we have now removed these legends.

- I found some petty typos:

Page 3: “explained” should read “explain”.

Page 4: “parts of heme molecule” should read “parts of the heme molecule”

Page 8: “for the numbers of” should read “for the number of”

Page 10: “measured features of Abs are include” should have either “are” or “include” removed.

Page 14: “rcontains” should read “contains”.

Page 16: “et al” should be followed by a period.

Page 17: “demonstrated that significant fraction” should read “demonstrated that a significant fraction”.

Page 20: “Adrich” should read “Aldrich”.

Page 32: “righ” should read “right”.

Page 32: “Graph Pad” should read “GraphPad”.

All indicated errors have been corrected.

Reviewer #2 (Remarks to the Author):

In their manuscript, Lecerf et al. show that a fraction of monoclonal therapeutic antibodies recognizes heme and that following the exposure to heme some of the therapeutic antibodies acquire antigen-binding promiscuity. In particular, they identify the amino acids in the antibody sequences most responsible for the interaction with heme. Considering the growing importance and interest in antibodies as therapeutics, and the fact that the development of therapeutic antibodies is challenging and time-consuming, being able to have strategies that quickly and effectively can predict early-stage different developability issues is of extreme interest.

The authors state that the non-specific binding of antibodies to heme can be used as a replacement for four commonly used assays to verify the developability of therapeutic candidates. Moreover, they address how the increased non-specificity of some therapeutic antibodies, when exposed to heme, can represent a risk for the use of such molecules, especially for the treatment of diseases where hemolytic conditions are present.

I do think that in the antibody engineering and development field having assays that can identify developability issues in very early phases is critical, not sure if the heme assay is ready to replace some of them or it's one that might be considered to be added.

I found interesting the increase of polyreactivity of some antibodies in the presence of heme, which I agree with the authors might be a critical issue for therapy of diseases where hemolysis might be present, although I don't think this aspect of the manuscript has been deeply explored. It would have been interesting to see how and if this polyreactivity plays a role in an *in vivo* or *ex vivo* model.

We are thankful to this Reviewer for critical assessment of our study. Indeed, impact on endogenous heme on the efficacy of therapeutic antibodies is of great importance. This is the reason that we have undertaken an independent research study on this topic. In 2010 J.A. McIntyre reported that certain therapeutic antibodies, which are approved for clinical use, acquire autoreactivity and polyreactivity upon exposure to heme (*Int J Cancer*. 2010, 127:491). Among these antibodies is Rituximab an anti-CD20 antibody used for treatment of certain forms of lymphoma. We performed *in vivo* experiments with lymphoma models to evaluate the impact of heme on therapeutic effect of Rituximab. Our preliminary data suggests that *pre*-exposure of the antibody to heme results in deterioration of its therapeutic effect in a model where human lymphoma cells were grafted in immunodeficient mice (see Figure below). In another experiment we used syngeneic model where mouse with complete immune system were grafted with mouse tumors expressing human CD20 and then treated with mouse anti-human CD20, which is also sensitive to heme. Induction of hemolysis in this mice caused impairment of anti-cancer therapy (see Figure bellow). However, we are not yet certain if this effect is caused by heme binding to to antibodies or alternatively acute intravascular hemolysis can induce systemic immunosuppression and thus blocking the anti-tumoral immune response by other mechanisms. Since there are many open questions and this is on-going topic of research in our laboratory we prefer not to present the effect of heme on the therapeutic efficacy of antibodies in the current manuscript but to present these data as an independent manuscript in the future.

Figure. Left panel: survival of mice grafted with human lymphoma cells and treated with native and heme-exposed Rituximab. Right panel: survival of mice grafted with mouse thymoma cells (EL4) transfected with human CD20 treated with anti-human CD20 IgG2a (CAT-13) or with isotype control. In indicated groups of mice intravascular hemolysis was induced by injection of phenylhydrazine.

Comments

Figure 1B: a) it would be interesting to show on the graphs the 44 antibodies among the 113 tested that are currently in clinical use, to clearly see if therapeutic antibodies behave better in a poly-specificity assay; b) in the results section (page 5 line 106) heme-binding antibodies were considered the one with a 10-fold higher intensity to immobilized heme as compared to the control protein alone. Why was the 10-fold chosen? Why in the figure a dashed line represents the 2-fold increased binding and not the 10-fold?

We are thankful to the Reviewer for this suggestion. We have now modified the figures and depicted distinguishingly antibodies that are approved for clinical use and those that are still in clinical trials. In addition, the threshold presented on the figures was modified to show 10-fold difference.

Figure 1D: in the text (page 6 line 111) it's reported that the antibodies recognizing heme with the highest intensity were subjected to additional analysis and the binding profile of 10 antibodies is shown in figure 1C. However, in figure 1D the sensorgrams of only fulranumab and figitumumab are shown, although in the figure legend tigatuzumab is also mentioned. I think it would be interesting to see the sensorgrams of all the 10 antibodies, or at least a table reporting the affinities measured for all 10 antibodies.

To address this issue, we performed additional kinetic measurements with SPR-based system. We tested the binding of heme to all 10 monoclonal antibodies that were pre-selected as top heme binders. The newly obtained data are presented as a novel supplementary figure. We included there the sensorogram of binding of heme to tigatuzumab. The text of the manuscript was edited to appropriately describe the new data.

Figure 2: similar to figure 1, it would be interesting to show on the graphs the 44 antibodies among the 113 tested that are currently in clinical use.

We have now modified this figure and depicted the clinically approved antibodies.

Figure 3: similar to figure 1, it would be interesting to show on the graphs the 44 antibodies

among the 113 tested that are currently in clinical use.

Although most of the antibodies have a binding signal below 0.5 absorbances before and after the exposure to heme, most of them do seem to have an increase in binding. For two of the tested targets, there are a few antibodies whose polyreactivity seems to be reduced after exposure with heme. Any explanation of those behaviors?

In the revised version of the manuscript we have now modified the figure to display the approved for clinical use antibodies.

We cannot provide unambiguous explanation for decreased of the reactivity of one of the antibodies after exposure to heme. But our hypothesis is that this antibody recognizes carbohydrate epitopes on the proteins. Our previous observations suggested that in contrast to protein antigens, where reactivity increases, the treatment of pooled human IgG with heme resulted in inhibition of binding to pneumococcal polysaccharide C. We hypothesize that heme competes with carbohydrate moieties for same binding site on antibodies.

Is there a correlation between non-specific binding to heme and the acquisition of antigen-binding polyreactivity? What is the percentage of the total antibodies that will be identified combining the two tests? Can the heme-induce-polyreactivity be correlated with “classical” developability assays?

The statistical analyses revealed that heme binding and heme induced polyreactivity correlate significantly see (Figure 7). Nevertheless, there are antibodies that bind heme without changes in polyreactivity reverse that acquire polyreactivity without showing considerable binding to immobilized heme on ELISA.

The heme-induced polyreactivity strongly correlated with developability tests that measure antibody hydrophobicity and cross-reactivity to other immunoglobulins. This data are presented on Figure 7.

Figure 4: the affinities for the most heme-reactive antibodies is between ~300-500 nM, why for this assay heme was used at 25-0.195 nM?

In this assay we tested actually dilutions (25-0.195 nM) of native and heme-exposed IgG molecules. The antibodies were exposed to a fixed concentration of heme (at 10 fold molar excess). The affinity of FcRn for human IgG1 is in range compatible with the range of concentrations of IgG1 (25-0.195 nM) used in our study.

To clarify this issue in the revised version of the manuscript we have now added two sentences in the results section.

Minor comments

Page 5 line 97: Heme is referred to as “Fe-protoporphyrin IX”, while in figure 1 legend as “Fe(II)protoporphyrin IX” and page 18 line 387 as “Fe(III)-protoporphyrin IX”. It would be better to have consistency in the nomenclature and to clarify if it is Fe(II) or Fe(III).

We agree that this lack of homogeneity is confusing. In the revised version of the manuscript this

issue was corrected. We also indicated that the study was performed only with oxidized form of heme.

Page 14 line 308: there is a typo “Abs rcontains”.

This typo was corrected.

REVIEWERS' COMMENTS:

Reviewer #1 (Remarks to the Author):

The authors have satisfactorily addressed my comments.
Despite some antibodies that bind heme may still be worth developing clinically, I think heme reactivity as shown in this work could be helpful to predict developability and help developers decide between (otherwise similar) antibody candidates.